# Hand Grasp Pose Prediction Based on Motion Prior Field

**DOI:** 10.3390/biomimetics8020250

**Published:** 2023-06-12

**Authors:** Xu Shi, Weichao Guo, Wei Xu, Xinjun Sheng

**Affiliations:** 1State Key Laboratory of Mechanical System and Vibration, School of Mechanical Engineering, Shanghai Jiao Tong University, Shanghai 200240, China; sjtu-shixu@sjtu.edu.cn (X.S.);; 2Meta Robotics Institute, Shanghai Jiao Tong University, Shanghai 200240, China

**Keywords:** grasp pose prediction, prior field, shared control

## Abstract

Shared control of bionic robot hands has recently attracted much research attention. However, few studies have performed predictive analysis for grasp pose, which is vital for the pre-shape planning of robotic wrists and hands. Aiming at shared control of dexterous hand grasp planning, this paper proposes a framework for grasp pose prediction based on the motion prior field. To map the hand–object pose to the final grasp pose, an object-centered motion prior field is established to learn the prediction model. The results of motion capture reconstruction show that, with the input of a 7-dimensional pose and cluster manifolds of dimension 100, the model performs best in terms of prediction accuracy (90.2%) and error distance (1.27 cm) in the sequence. The model makes correct predictions in the first 50% of the sequence during hand approach to the object. The outcomes of this study enable prediction of the grasp pose in advance as the hand approaches the object, which is very important for enabling the shared control of bionic and prosthetic hands.

## 1. Introduction

Bionic robot hands play an important role in human–machine collaboration and rehabilitation engineering to improve the life quality of amputees. During the process of grasping an object by a prosthetic hand, the finger and wrist joints require dexterous control. Nevertheless, the conventional myoelectric control performance, which relies on the nervous system of the amputation stump, is inadequate to cope with the manipulation requirements of prostheses [1]. To reduce the burden on the user, machine intelligence is needed to help the human with delicate control according to human intent. In recent years, there has been a growing trend towards introducing a shared control framework for the control of multi-degree-of-freedom prosthetic hands. Many studies have involved attachment of a camera to the prosthetic hand to capture objects and environmental information to assist the user with pre-shape adjustment of the prosthetic hand or control of the wrist joint. The authors of [2,3] installed a depth camera on the wrist to capture object information to realize the adjustment of the wrist angle and the selection of pre-grasp gestures. Other studies, such as [4,5,6,7], involved installation of RGB cameras on prosthetic hands, relying only on RGB images to achieve the same results.

There are several shortcomings in the current approach to machine-vision-based hand-sharing control of prostheses. First, current research does not consider the prediction of grasp pose. In [3,6], for example, the essence of the work is consideration of the object–gesture classification problem, which requires the user to set up the appropriate approach posture during the pre-shape process. The aim is only to grasp, but there is no consideration of whether the different contact areas and different approach directions are compatible with everyday human habits. Additionally, the pre-shape performance mainly depends on the camera capture perspective. In [3,4], a predictive model is used to learn directly from the captured image, making the application highly viewpoint-dependent and, thus, difficult to generalize to multi-view deployments.

In order to address the abovementioned requirements, inspiration was needed from several cross-cutting areas, including human motion analysis and robot grasping planning. In studies related to the analysis of human motion analysis, researchers have undertaken a considerable amount of research on data collection and analysis. In [8,9], human upper limbs were modeled with different degrees of freedom for daily activities and clustering analysis was performed. In [10,11], whole-body motion data for a human during grasping action were collected, and then a generative model was trained to produce a sequence of grasping motions for new objects. In [12], similarly, data for the grasping behaviors were collected and a dataset was produced. The authors of [13,14,15] collected and analyzed data for the contact area and hand posture during the grasping of objects. In [16], different grasping poses of objects were generated for everyday scenes, while, in [17], future human motion was predicted with a known human posture and environment. Much work has also been carried out by many researchers in the field of robotics on the planning and prediction of grasping poses. The authors of [18] generated various grasping poses of objects using cVAE for a two-finger gripper robot arm. In two studies [19,20], an implicit neural field was created centered on the object that encoded the distance from the spatial point to the grasping pose into the field, while the authors of [21,22] generated dexterous grasping poses and grasping gestures using generative models.

However, assessment of research advances in human motion analysis and robot grasping planning suggests that the fields of motion behaviour analysis and robot grasp pose prediction are not yet well integrated. Furthermore, there is a lack of grasp pose prediction planning in the field of the shared control of prosthetic hands based on human behaviour analysis as a starting point. Therefore, we propose a grasping pose prediction method based on a motion prior model to achieve grasping pose prediction for multiple grasping gestures of objects and arbitrary approach directions, providing a theoretical basis for semi-autonomous control of the prosthetic hand-wrist. The core idea of this paper is to build an object-centred prior field of human grasping trajectories, to map each trajectory point to a final grasping pose, and to produce a model that maps every point in space to a certain grasping pose.

To overcome the challenges, it is of great importance to explore a predictive model oriented towards the shared control of prosthetic hands, which is capable of predicting pre-shape gestures and grasping postures in advance, laying the foundation for joint control. In summary, we propose a new grasp pose prediction framework for prosthetic hand shared control based on a motion prior field. We establish an object-centered motion prior field of grasp motion trajectories and create a prediction model that predicts grasp poses in advance, covering arbitrary approach directions and multiple pre-shape types. This avoids the dependence of similar work on the camera capture perspective. Note that, we define the grasp pose as consisting of two parts: one is the 6-DOF wrist pose in the object coordinate system, and the other is the pre-shape type. Among them, the definition of pose of the former is the same as that of the end claw of the general mechanical arm, while the latter is aimed at grasping prosthetic hands. In the following sections, we explain the hardware and the testbed that we use in Section 2. This is followed by a detailed introduction to the motion prior field, provided in Section 2.2.1, Section 2.2.2, Section 2.2.3 and Section 2.2.4. Finally, we analyse the comparative results under different design parameters in Section 3, including the prediction accuracy for the motion prior field and the predicted grasp pose error in the sequence.

## 2. Materials and Methods

We established an object-centered motion prior field, and used each spatial point in this field to represent the pose of the hand relative to the object. Therefore, in order to integrate human motion habits as prior information into the prior field, we collected a large number of motion sequences of human hand approaches and grasps. The sequences obtained contained two pieces of information: the hand pose in the trajectory and the hand pose when grasping. We sought to produce a mapping model based on these collected data, which would map the hand pose in the trajectory to the hand pose when grasping. With this mapping relationship, any point in space can be regarded as a hand pose in the trajectory. Therefore, we can predict the pose of the hand when it finally grasps an object before it even touches it.

As shown in Figure 1, this work is divided into two main sections: hand–object localisation and grasp pose prediction. We aim to obtain the relative hand–object pose $T_{\mathrm{object}}^{\mathrm{hand}}$ by using a motion capture system before feeding this pose into the learned prediction model MPFNet and, finally, outputting the predicted grasp pose $P\left(g_{\mathrm{object}}^{\mathrm{hand}}|T_{\mathrm{object}}^{\mathrm{hand}}\right)$. The above process is performed while the hand is still at a certain distance from the object, with the intention of predicting the possible grasping poses in advance and laying the foundation for later wrist joint control.

In order to achieve this, we capture the motion sequence information through the motion capture device, establish the a priori field of the object-centred grasping motion trajectory, and design the prediction model MPFNet to map any hand. The second step is to establish an a priori field of object-centred grasping motion trajectories by capturing motion sequence information with the motion capture device, and to design a prediction model MPFNet to map arbitrary hand poses in space to grasping poses.

### 2.1. Hardware and Test Bed

The core step is the construction of an a priori field of motion trajectories, which requires the accurate capture of hand and object poses throughout the approach–grasp process. As shown in Figure 2, we prepared 19 types of objects for daily use, and placed 8 motion capture devices evenly around the table. Eight OptiTrack (NaturalPoint, Inc., Corvallis, OR, USA) motion capture cameras were used, arranged in four corners, with an acquisition period of 70 µs; four 9 mm diameter markers were installed on the hand and several 3 mm diameter markers were installed on the object. For the hand pose tracking, we used a glove and installed two markers at the wrist and palm sections, respectively, to create a rigid body model of the hand. Note that, for hand pose tracking, we treated the wrist pose as the overall hand pose and ignored the specific posture of each finger, as the wrist pose is more important for predicting the grasp pose, while the finger posture configuration can be determined by the pre-shape gesture.

### 2.2. Motion Prior Field

In this work, we define the grasp pose as consisting of two parts: **pre-shape type** and **wrist 6D pose**, as shown in Figure 3. Since our future work is mainly oriented towards shared control of prosthetic hands, and the fingers of prosthetic hands are much less dexterous than human fingers, we do not include the finger posture as part of the grasp pose, but use the more general pre-shape type instead of the joint configuration of the fingers. Specifically, our choice of pre-shape type is derived from the taxonomy criteria in [23], where two to three pre-shape types can be artificially classified according to the different grasping parts of each class of objects, designated as *c*, which is a discrete variable. The wrist 6D pose $g_{\mathrm{object}}^{\mathrm{hand}}$ is a Euclidean transformation that represents the wrist 6D pose when the hand is in contact with the object in the object coordinate system and is a continuous variable in Euclidean space.

#### 2.2.1. Data Collection

In order to cover as many objects as possible used by ordinary people in daily life, while taking into account a standard benchmark, we chose 19 objects from YCB-video [24], which are all standard parts of a YCB object [25] (http://ycbbenchmarks.org/, accessed on 10 June 2023). For each of these objects, the subjects grasp each object with two or three pre-shape types, respectively, and cover the approach direction and the approach angle as much as possible.

In order to meet the requirement that the acquisition process covers all the space in which the human upper arm moves, we followed the principle of [26] for object placement, as shown in Figure 4. The rays start at 0° and go to 160° with 20° intervals. In the direction of each ray, the objects are located at two distances, r1 = 30 mm and r2 = 60 mm. Here, we discarded the leftmost ray because humans normally do not grasp objects at this angular range. We placed the object at two positions on each ray, one near and one far, to cover as much of the grasping motion trajectory as possible. Eight subjects (six males and two females) participated in the data collection process, and an average of 500 grasps were performed per person.

In addition to placing the objects in different positions, we also rotated the objects themselves. At each placement position, we rotated the objects at random angles for multiple grasps. For each object, the subjects covered as many approach directions and approach angles as possible; thus, we were able to construct a variety of grasping poses (including various pre-shape types and a large number of wrist 6D poses) for the object over a 360° spatial range. We performed a large number of grasps for 19 objects in all directions and for various pre-shape types, and obtained a total of about 4200 accurate grasp poses, of which each pre-shape type for each type of object corresponded to 100 wrist 6D poses.

After completing each “approach–grasp” motion sequence, we post-processed the motion trajectory corresponding to each grasp pose. The purpose of post-processing was as follows:The initial hand pose was the same at the beginning of each motion capture, so we excluded the first 10% of the motion sequence;The purpose of our modeling was to help the system complete the prediction when the hand was far away from the object, so trajectory points too close to the grasp position were less valuable; hence, we excluded the last 30% of the data for the motion sequence;Since the hand did not move uniformly during the approach, and the acquisition frequency of the motion capture device was fixed, the collected trajectory points were not uniformly distributed in the sequence; thus, we interpolated the whole segment of data to ensure that the data tended to be uniformly distributed in space.

Finally, each grasping pose corresponded, on average, to 50 trajectory points.

#### 2.2.2. Constructing the Motion Prior Field

In order to establish the object-centered motion trajectory prior field, it is necessary to transform the hand pose at each moment of the “approach–grasp” process to the object coordinate system. Let the pose of the hand and object in the motion capture system coordinate system (world coordinate system) be $T_{\mathrm{world}}^{\mathrm{hand}}$ and $T_{\mathrm{world}}^{\mathrm{object}}$, respectively, then the transformation relationship is as follows: (1)$$\begin{array}{c} T_{\mathrm{object}}^{\mathrm{hand}}=T_{\mathrm{object}}^{\mathrm{world}}*T_{\mathrm{world}}^{\mathrm{hand}}\\ ={\left(T_{\mathrm{world}}^{\mathrm{object}}\right)}^{-1}*T_{\mathrm{world}}^{\mathrm{hand}}\\ =\left[\begin{array}{cc} {\left(R_{\mathrm{world}}^{\mathrm{object}}\right)}^{T} & -{\left(R_{\mathrm{world}}^{\mathrm{object}}\right)}^{T}*t_{\mathrm{world}}^{\mathrm{object}}\\ 0^{T} & 1 \end{array}\right]*\left[\begin{array}{cc} R_{\mathrm{world}}^{\mathrm{hand}} & t_{\mathrm{world}}^{\mathrm{hand}}\\ 0^{T} & 1 \end{array}\right]\\ =\left[\begin{array}{cc} {\left(R_{\mathrm{world}}^{\mathrm{object}}\right)}^{T}*R_{\mathrm{world}}^{\mathrm{hand}} & {\left(R_{\mathrm{world}}^{\mathrm{object}}\right)}^{T}*\left(t_{\mathrm{world}}^{\mathrm{hand}}-t_{\mathrm{world}}^{\mathrm{object}}\right)\\ 0^{T} & 1 \end{array}\right] \end{array}$$

Figure 5 shows the result of a prior field construction for a part of all the objects: one color represents a pre-shape type, and one to two samples of each pre-shape type are selected as shown in the figure. In the scatterplot on the right, each point represents the trajectory point of the hand motion corresponding to one specific grasp pose, and all the trajectory points are collected to construct the motion prior field.

As a result, we map each trajectory point in space to a specific grasp pose. In the motion prior field, each point can be represented as a vector: (2)$$p=\left(q_{x},q_{y},q_{z},q_{w},t_{x},t_{y},t_{z},c,g\right)$$

In this, $\left(q_{x},q_{y},q_{z},q_{w}\right)$ denote the current rotation information of the hand and $\left(t_{x},t_{y},t_{z}\right)$ denote the translation information of the hand. *c* and *g* represent information about the grasp pose to which this point is mapped, i.e., the pre-shape type *c*, and the wrist pose $g_{\mathrm{hand}}^{\mathrm{object}}$. Next, before training the prediction model, we need to consider how to pre-process such a huge amount of sampled data. One idea is to cluster the wrist 6D poses.

Since each pre-shape type will correspond to a large number of wrist 6D poses, and these wrist poses are uniformly distributed in space in all directions, we have two ways to deal with the wrist poses: the first is to cluster them to obtain a number of averaged wrist poses; the second is not to cluster them and to treat each wrist posture as a separate individual. Figure 6a shows the result of clustering; the middle layer corresponds to the averaged cluster manifolds. It is reflected in the motion prior field as a “piece” of trajectory points mapped to an average pose, which makes the field more linearly separable in space. Figure 6b shows the result without clustering, which is reflected in the motion prior field, making the field less linearly separable in space. It is worth noting that, as the number of cluster manifolds increases, Figure 6a converges to Figure 6b, because, if the number of cluster manifolds is equal to the number of grasp pose individuals, then there is no clustering, and the number of cluster manifolds is 100. We explored the effect of the cluster manifolds on the prediction of the model in our experiments.

As the online process is bound to have errors in the 6D pose estimation of the object based on the RGB camera, this will cause the reconstructed hand pose $T_{\mathrm{object}}^{\mathrm{hand}}$ to have deviations of 1–5 cm for both rotation and translation compared to the real value. These deviations will make it difficult for the prediction model to converge. Therefore, the robustness of the system to the pose estimation errors needs to be improved by adding noise. At the sampling frequency of the motion capture system, the spatial separation of adjacent trajectory points during the normal approach of the human hand to the target is about 2 cm, so we add N(μ=0,std=0.1) and N(μ=0,std=0.01) Gaussian noise to the rotation of the wrist and the translation of the wrist in the trajectory points, respectively.

#### 2.2.3. MPFNet

Based on the constructed a priori field, we can know the grasping pose (i.e., pre-shape type + wrist pose) corresponding to these existing trajectory points in space. However, for newly collected trajectory points, the system also needs to make a prediction of its corresponding grasping pose. Therefore, we consider designing a classification model to learn a nonlinear mapping relationship $P\left(g_{\mathrm{object}}^{\mathrm{hand}}|T_{\mathrm{object}}^{\mathrm{hand}}\right)$ from the trajectory points to the grasp poses.

We used a simple multi-layer perception with five layers and undertook an experimental comparison of different input and output cases, as Figure 7 shows. First, there are two choices of inputs: 6D pose and 3D position. The former is represented by a 7-dimensional vector (quaternion + translation) and the latter by a 3D vector (translation). The dimension of the output is influenced by the cluster manifolds, which have a scale of $num_{manifolds}*num_{gtype}$, representing the average number of grasp poses clustered out in total. $num_{manifolds}$ represents the number of clustered manifolds within each pre-shape type, and $num_{gtype}$ is the amount of pre-shape types. Therefore, the labels are also in the form of a one-hot encoding, representing one of the average poses of the cluster manifolds. In order to converge the model, and to make the incorrect prediction results as close as possible to the correct ones, we use a cross-entropy loss function based on softmax as follows: (3)$$L=-\frac{1}{N}\sum_{i}\sum_{c=1}^{M}y_{ic}\log\left(p_{ic}\right),p_{ic}=\frac{e^{z_{c}}}{\sum_{j=1}^{M}e^{z_{j}}}$$

*M* represents the total number of categories of clustered manifolds, which is the dimension of the output. $z_{j}$ represents the last output of the full connection layer, where $z_{c}$ represents the output when the class is c. *N* denotes the number of training samples, and $y_{ic}$ and $p_{ic}$ represent the probabilities of ground truth and prediction, respectively.

The complete process of prediction is as follows: First, we place an object in a certain location. A second person standing in front of the object moves their hand towards the object. Then the position and orientation of the hand and object are measured. After coordinate transformation, we can obtain the hand pose in the object coordinate system. Finally, the hand pose $T_{\mathrm{object}}^{\mathrm{hand}}$ is inputted into MPFNet to obtain the predicted grass pose $g_{\mathrm{object}}^{\mathrm{hand}}$.

#### 2.2.4. Evaluation Metrics

The benchmark used was based on human performance, because our evaluation criteria were all based on the dataset of human hand grasping objects. We defined two evaluation metrics for the prediction model:Prediction (classification) accuracy. This represents the classification accuracy of all trajectory points in the motion prior field and is able to measure the fit of the prediction model to the whole field;The predicted grasp pose error in the sequence. This focuses on the 50 percent of the trajectory points of one sequence of approach. Each trajectory point generates a predicted grasp pose; this grasp pose will have some error (or no error) with the real pose before. We record the errors in this approach sequence:
(4)$${err}_{\mathrm{dist}}=\frac{1}{m}\sum_{x\in X}∥\left({\hat{R}}_{i}x+{\hat{t}}_{i}\right)-\left(R_{i}x+t_{i}\right)∥$$We use the distance between the predicted pose and the real pose to represent the error size. *x* are the selected points to represent the hand, and *X* is the set of *x*, with *m* representing the amount of *x*. As shown in Figure 8, the green hand represents the true grasp pose of this approach sequence; the gray hand represents the incorrectly predicted grasp pose at some point.If the errors are accumulated for each moment:
(5)$${Err}_{dist}=\sum_{i=0}^{t-1}err_{dist}$$
then the total error accumulation ${Err}_{dist}$ of this approach sequence will be obtained. The smaller this accumulation is, the more stable and reliable the prediction model is during one approach; however, if this accumulation is larger, the prediction model is more unstable and less reliable.

## 3. Results

The purpose of our experiments was to explore the effect of the input-output form on the model performance. To be more specific: we used different representations of trajectory points (7-dimensional pose or 3-dimensional position), and different cluster manifolds (we chose 4, 20, 40, 60, 80, and 100 as representative manifolds). Since, in the future, the model will need to contribute to the control of the prosthetic wrist joint, the number of manifolds cannot be too small, otherwise the averaged grasping pose would be sparse, which would be detrimental for wrist joint control. Therefore, we set the minimum number of manifolds to four. In this way, one pre-shape type within 360° will cluster four average wrist poses with a 90° adjacent interval, which is the maximum acceptable adjustment range for the wrist and forearm [26]. Next, we compared and analyzed two aspects: the prediction accuracy among the motion prior fields, and the predicted grasp pose error in the sequence.

### 3.1. Prediction Accuracy among the Motion Prior Fields

As shown in Figure 9, the horizontal coordinates represent the different cluster manifolds, where 100 manifolds means that each pose is considered as one manifold, which corresponds to the case of "no clustering". The blue box and the orange box indicate the results when the input is a 7-dimensional pose and a 3-dimensional position, respectively. It can be seen that the model predicts better with pose than with position, regardless of the clustering. When the input is pose, the prediction accuracy decreases and then increases with increase in manifolds. In particular, when the manifolds are 100, the accuracy is almost the same as that when the manifolds are 4. Generally speaking, the fewer the manifolds, the better the linear differentiability of the field. However, we find that the linear differentiability of manifolds and fields is not simply negatively correlated. Therefore, when the manifolds are in the middle (e.g., 40, 60), the model is artificially reduced in dimensions, which makes the fit of the neural network constrained, so the accuracy will be reduced. When there are few or many manifolds (e.g., 4 and 100), the field is closer to the original form of the data, i.e., either the simple divisible space at 4 manifolds or the original high-dimensional random distribution at 100 manifolds. Therefore, this is when the neural network has less constraint in fitting and has higher accuracy.

### 3.2. Predicted Grasp Pose Error in Sequence

We also compared the two input cases and different numbers of cluster manifolds for the predicted grasp pose error in the sequence, as shown in Figure 10. The horizontal coordinate indicates the moment of motion and the vertical coordinate indicates the error distance from the predicted grasp pose to the real grasp pose at the current moment. Figure 10a shows the results with the input as pose. It can be seen that the error distance shrinks faster and the overall error decreases as the manifolds increase. This is because the smaller the manifolds are, the fewer the average grasp poses will be, and then each prediction will have a “steady-state error” with the real grasp pose. Figure 10b shows the result when the input is position—the error is larger and the variance is greater than when the input is pose. In summary, with pose and 100 manifolds, the error distance ends up being about 1 cm, which is sufficient for wrist control.

The error distance at each moment in the sequence is summed up to obtain the bar chart in Figure 11. In the experiment, our sampling frequency was 50 Hz, and we intercepted the first 50% of the sequence in the hand approach process, corresponding to about 1000 ms in time. It can be seen that the error distance summation for the input pose was lower than for the input position in any of the cluster manifolds. In addition, the error distance summation was minimal for cluster manifolds of 100 (i.e., no clustering). This suggests that the finer the grasp pose is divided, the better the model performs in sequence prediction.

## 4. Discussion

For better understanding, we visualised and analysed the predictions of the model throughout the ‘hand approaching object’ process. As shown in Figure 12, the left column represents the four key frames during the hand approaching the object, and the three columns on the right show the predictions of the model under different situations. The flesh-colored hand shows the reconstructed hand pose (historical data), green represents the real grasp pose, and gray represents the incorrectly predicted grasp pose. The second column is the result of the motion capture reconstruction. In particular, in the second key frame, the grasp pose is spatially close to the true grasp pose, although it is incorrectly predicted. and we can see that the hand pose is tracked very smoothly and the correct prediction is made at the third key frame. In addition, we explored the robustness of the model to the inputs. We added Gaussian noise to the pose of the hand in the sequence, as shown in the third column. It can be seen that the hand motion sequence was no longer smooth. However, by training the model on the motion prior field after adding the noise, we were still able to obtain accurate predictions. In the third key frame, although the predicted grasp pose was wrong, it was very close to the real grasp pose (green hand). In the fourth key frame, the system’s prediction was also completely correct. Therefore, the model was able to make accurate predictions in the first 50% of sequence of the hand approaching the object.

Based on the experimental results, we found that, by encoding human motion trajectories into fields as a priori information, we were able to obtain a mapping from any point in space to the final grasp pose, and, therefore, a framework for prediction could be constructed. After comparison of experiments on different objects, we believe that neither weight nor size plays a major role in the prediction of grasp pose, but, for the shape and pre-shape category of the object, as long as the shape of the object is similar, the prediction model is supported. To the best of our knowledge, this paper is the first to use prior knowledge to predict the manual grasp pose. In the shared control of grasp in a prosthetic hand, the results of this study can be used as input information to the control end, i.e., the grasp pose is predicted in advance and used as a control target for the wrist joint and the hand pre-shape. In addition, the results of this paper can also be useful in scenarios such as human–robot handover, for example, by predicting the grasp area of the human hand in advance and, thus, helping the robot arm to plan the grasp area. Finally, there are still some shortcomings and improvements that require to be made. First, the variety of objects collected in this paper was limited and did not cover all scenes from everyday activities. Second, the prediction model is essentially a classification after dense sampling and does not generate a completely new grasp pose at each capture, so it has limited robustness and is difficult to migrate. Moreover, the purpose of designing the prediction model was to predict the final grasp pose as accurately as possible in the whole approach process, so we did not discuss which state is the most suitable for prediction in this process. However, we will investigate these issues further in follow-up work.

## 5. Conclusions

In this paper, focusing on the shared control of prostheses, we proposed a framework for grasp pose prediction based on hand–object localization and a motion prior field. We collected a large amount of “approach–grasp” motion data to build an object-centered motion prior field. We related the trajectory points in space to the grasp pose and trained a prediction model to map this correspondence. We experimentally investigated the effect of different input dimensions and different cluster manifolds on the prediction effect of the model. The results showed that the model performed best in terms of prediction accuracy and error distance in the sequence when the input was a 7-dimensional pose and the cluster manifolds were 100 (i.e., no clustering of the grasp pose). The model enabled the system to accurately predict the final grasp pose in the first 50% of the grasp motion sequence. Our results represent grasp pose predictions based on hand–object localization, which essentially map the state of the end effector in SE(3) space to the final grasping state. Moreover, the prediction model is very simple, so other researchers can refer to our ideas and potentially use the network model we provide directly to quickly predict grasp pose and reduce the burden of grasping planning. In the future, we intend to apply the current prediction model to the semi-autonomous control of a 2 DoF prosthetic hand–wrist to accomplish grasping of everyday objects in different directions and for different contact areas.

## Figures and Tables

**Figure 1 biomimetics-08-00250-f001:**
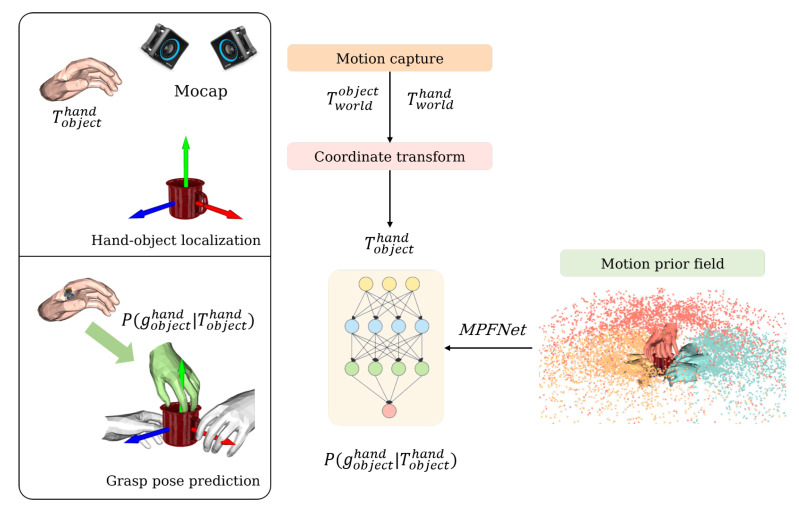
The pipeline of the framework. The motion capture system reconstructs the hand–object-pose $T_{\mathrm{object}}^{\mathrm{hand}}$, and then the model predicts the final grasp pose $P\left(g_{\mathrm{object}}^{\mathrm{hand}}|T_{\mathrm{object}}^{\mathrm{hand}}\right)$ based on $T_{\mathrm{object}}^{\mathrm{hand}}$. The green hand in the lower left figure represents the correct prediction.

**Figure 2 biomimetics-08-00250-f002:**
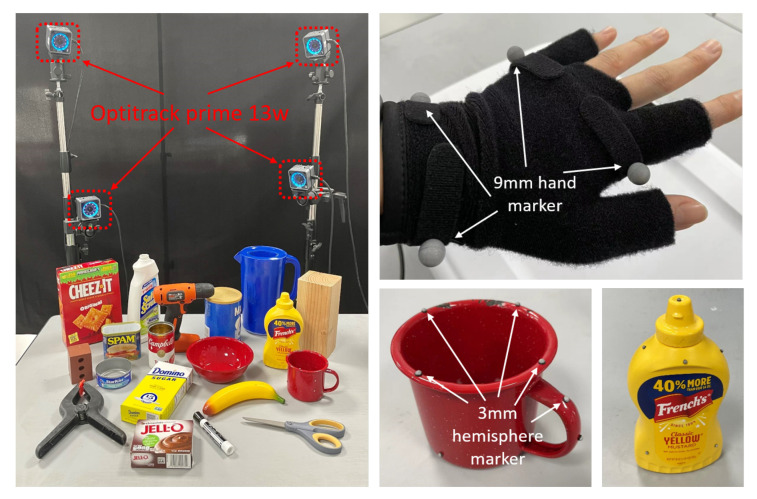
Devices used for data acquisition and experiments.

**Figure 3 biomimetics-08-00250-f003:**
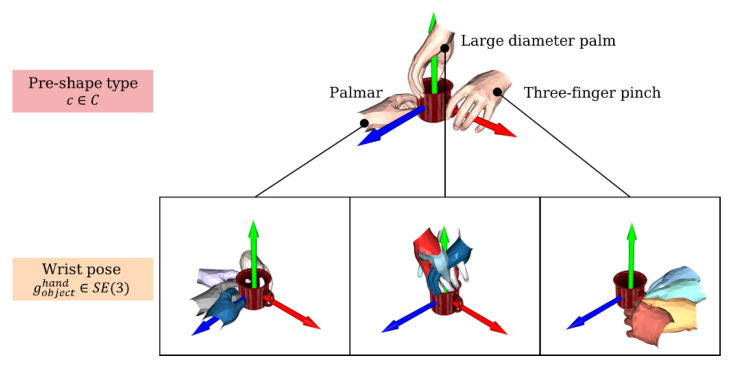
Definition of grasp pose, consisting of two parts: pre-shape type and wrist pose.

**Figure 4 biomimetics-08-00250-f004:**
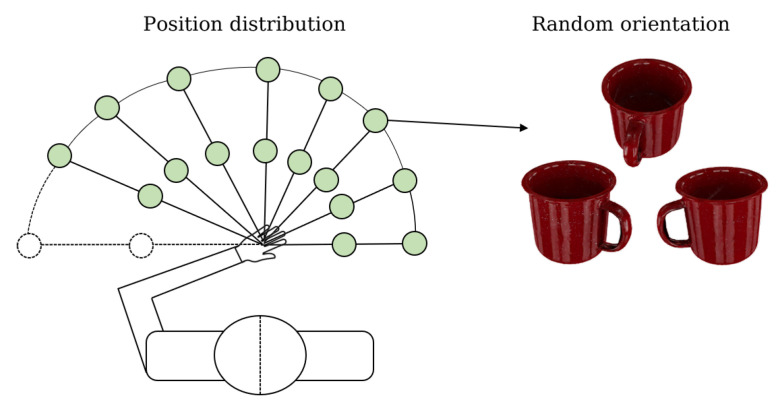
Object placement plane top view.

**Figure 5 biomimetics-08-00250-f005:**
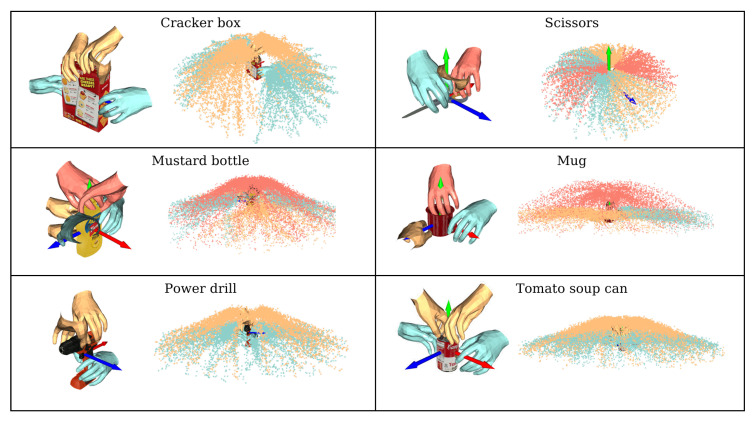
Motion prior field diagram of some objects. The color of the scatter corresponds to the pre-shape type. Note that, this is an object-centered representation, so the axes belong to the object coordinate system.

**Figure 6 biomimetics-08-00250-f006:**
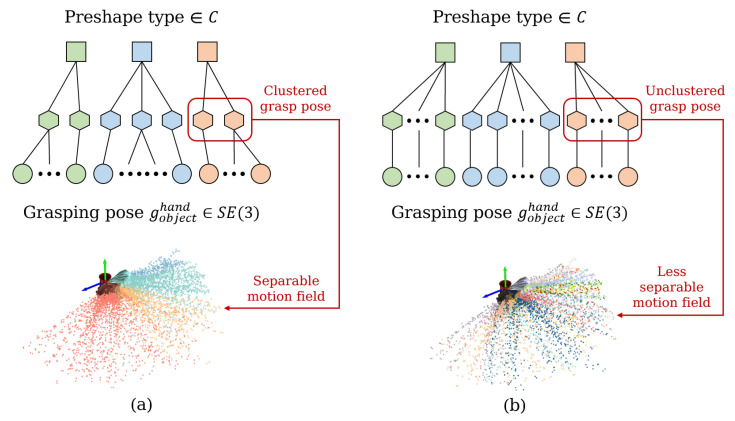
Clustering of grasp pose. (**a**) cluster manifolds < total number of grasp pose; (**b**) cluster manifolds = total number of grasp poses, which is equivalent to no clustering.

**Figure 7 biomimetics-08-00250-f007:**
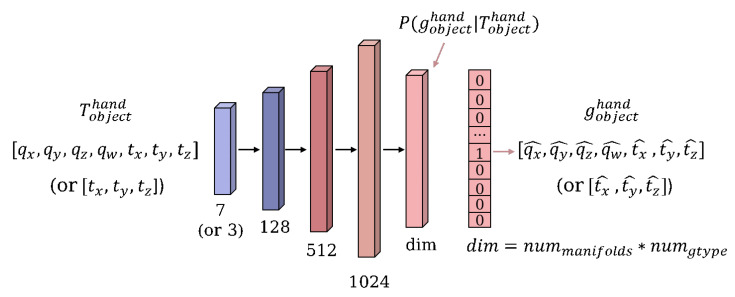
Structure of multi-layer perceptron (MLP). The numbers, such as 128, 512 and 1024, are the dimensions of the fully connected layer.

**Figure 8 biomimetics-08-00250-f008:**
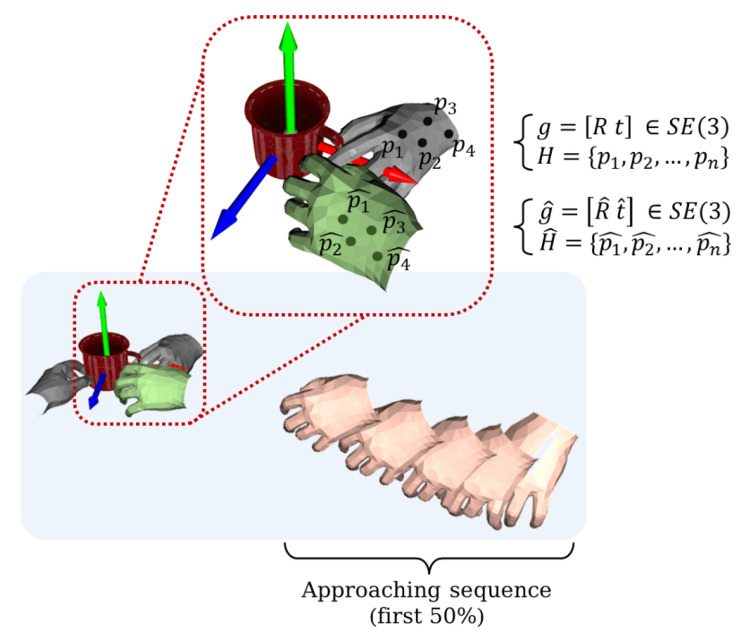
Predicted grasp pose error in the sequence. The flesh-colored hand represents the true grasp pose and the gray hand represents the incorrectly predicted grasp pose.

**Figure 9 biomimetics-08-00250-f009:**
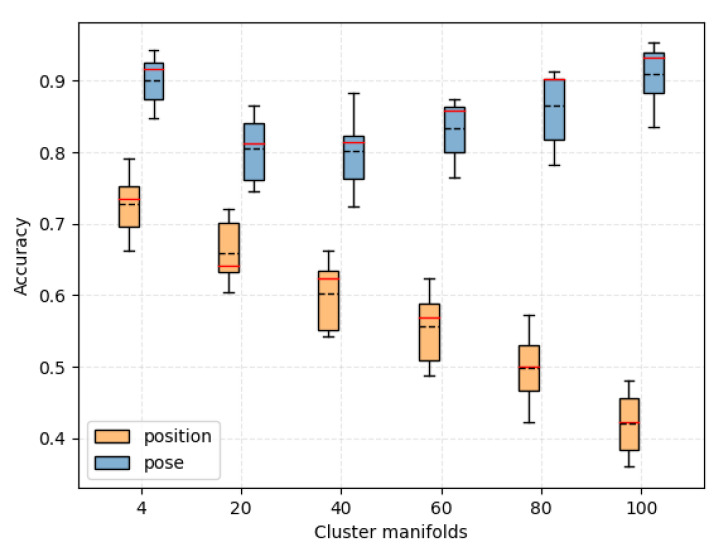
Prediction accuracy among motion prior fields.

**Figure 10 biomimetics-08-00250-f010:**
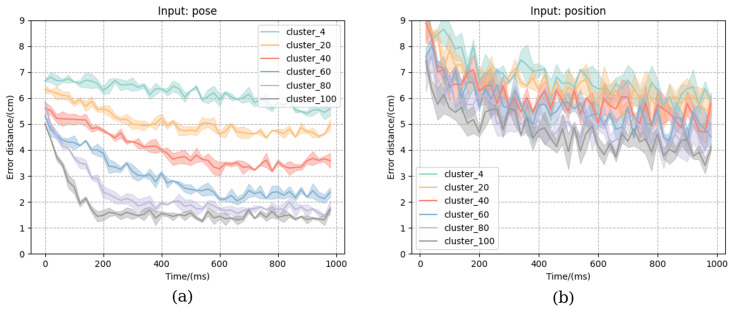
Predicted grasp pose error in sequence. (**a**) and (**b**) respectively show the change of the error distance of the prediction model within 1000 ms when the input is pose and position.

**Figure 11 biomimetics-08-00250-f011:**
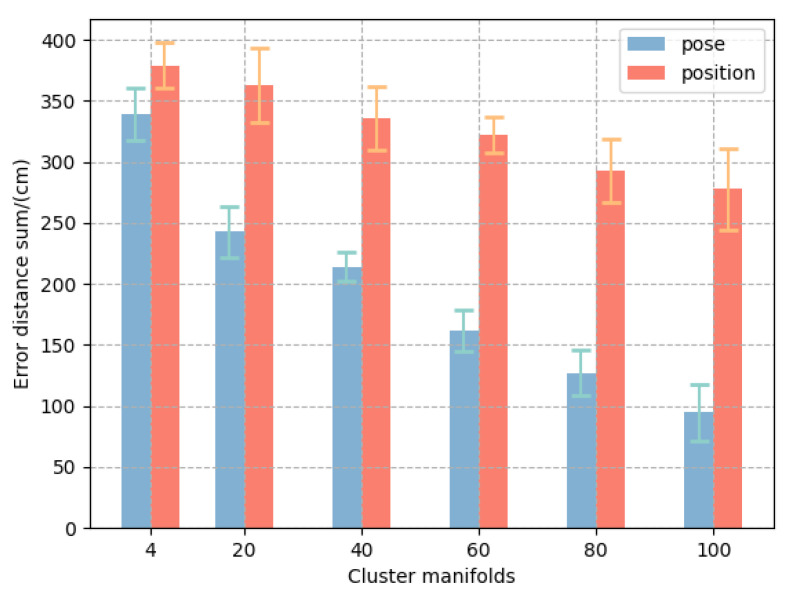
Summation of predicted grasp pose error in sequence.

**Figure 12 biomimetics-08-00250-f012:**
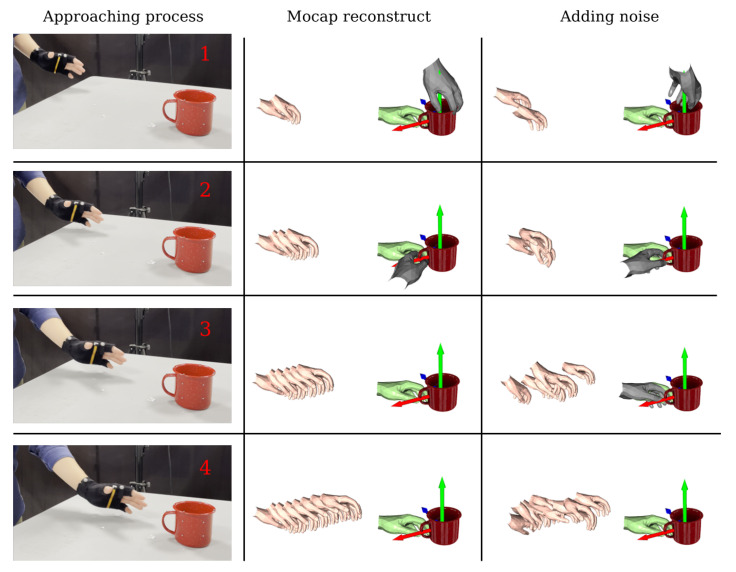
Demonstration of predictive model performance during hand approach to an object. The left column represents the four key frames during the hand approaching the object, and the right column is the result of the motion capture reconstruction.

## Data Availability

The data and code during the current study can be obtained from the author upon reasonable request.

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
