# Peer review of "Hand Grasp Pose Prediction Based on Motion Prior Field"

_biomimetics, 2023, doi:10.3390/biomimetics8020250_

Round 1

Reviewer 1 Report

The paper proposes a framework for predicting grasp pose, aiming to facilitate shared control of dexterous hand grasp planning. By establishing an object-centered motion prior field, the model accurately predicts grasp poses during the hand's approach to the object. Results show high prediction accuracy and low error distance, with correct predictions made in the initial 50% of the sequence. Work is interesting. However, there are some minor issues need to be considered.

 1. In the paper, it is preferable to provide the key parameters of the target object, such as dimensions, weight, and so on.

 2. Can the author specify which parameters of the target object have a major impact on the predicted grasp pose? For example, for the same object, is it the weight or the size that has a greater influence on the predicted grasp pose? It helps the scalability of the proposed method.

Minor editing of English language required

Reviewer 2 Report

Dear authors,

The work is aimed at planning grasping motion planning, which is an important problem in robotics. My main recommendations are related to improving the presentation, clearly defining the experiment and interpreting the results.

Here are my specific suggestions:

1.     Is it good to clarify what is meant by shared control in the summary? Between which elements?

2.     2. In the Introduction, define (or cite) the term "grasping pose"? You are probably referring to the position and orientation of the palm. What elements define the pose in this study and how does it differ with a robotic arm configuration?

3.     I recommend that the purpose of the research be written at the end of the Introduction.

4.     Explain what you think is the correct posе for grasping a certain object and why?

5.     Line 119, here you indeed define the pose as "wrist 6D pose" but it is still good to be specific - 3 components for position and 3 for orientation. This will be better clarifyed if you have a coordinate system drawn in Fig. 3 that is connected to the palm.

6.     I think it's good to draw the placement of the three coordinate systems "world", "hand" and "object". Perhaps in figure 5 or a new figure.

7.     More clearly describe what the initial conditions of the experiment were. Fig.4 could be improved: What is the initial configuration of the hand? Is it always the same? Specify target dimensions (polar coordinates that change). What is a good posture for grasping a particular object? A cup can be grasped by a person in many different ways. Which one do you think is correct?

8.     I think formula 1 needs clarification. What are the transformation matrices T, these must be homogeneous transformation matrices with dimension 4x4. R - 3x3 rotation matrices, t – sets a three-coordinate position, O - 1x3 null block. Then please check the second line of formula 1 if it is correctly written?

9.     Formula 3 - it is not clear what zc and zj are?

10.  What and how is predicted should be more clearly described.

First we place an object in a certain location. A second person positioned in a certain way moves his hand towards the object. Third, position and orientation are measured...

What is compared as a benchmark to make a prediction about its performance? Human or virtual model performance?

11.  Maybe Fig.12 should be in the results section.

12.  Analysis of results. It is not clear which initial hand configuration (or previous movement) is best for grasping a particular object (e.g. a cup)?

13.  In conclusion, it is good to indicate how your results can be used by other researchers for motion planning of a prosthetic or robotic arm.

14.  References [24] where is it published?

Best regards,

Reviewer

Reviewer 3 Report

1. Is the sentence on lines 35-37: “[4] and [3] learns a predictive model directly from the learning a predictive model directly from the captured image” correct? Please check.

2. When collecting grasping motion trajectory data, does the moving speed of human hands have an impact on the data collection results due to the different placement distances of different objects? If so, how to eliminate it?

3. What is the final expression for the nonlinear mapping relationship P (g, T) mentioned in section 2.2.3 ? What is the relationship with formula (3)?

4. What is the relationship between the layers in the multi-layer perceptron (MLP) shown in Figure 7 ? How to obtain P by gradually passing through each layer from T ?

5. What is the meaning of the numbers: 128, 512, 1024 in Figure 7 ?

6. Limitations of the current work should be added to the manuscript.

Round 2

Reviewer 2 Report

Dear authors,

Thanks to the authors for the corrections and answers to my questions. I think there is work being done and it may evolve to more real applications in the future.

I think the text can still be improved in the direction of clearer presentation. For example:

In Fig.4 it is not correct to say that the positions of the objects are "random". It is obvious that they are placed in 20 degrees. I think it should be specified that the rays start at 0 degrees and go to 160 degrees? In the direction of each ray, the objects are located at two distances r1=x[mm] and r2=y[mm]? Check if it is also true for the orientation?

I ask the authors to revise the text to improve the presentation.

Best regards,

Reviewer

Author Response

Thanks for your adivice, and please see the attachment.
